# Catalytic asymmetric radical aminoperfluoro-alkylation and aminodifluoromethylation of alkenes to versatile enantioenriched-fluoroalkyl amines

Jin-Shun Lin[1],*, Fu-Li Wang[1],*, Xiao-Yang Dong[1],*, Wei-Wei He[1], Yue Yuan[1], Su Chen[1] & Xin-Yuan Liu[1]

Although great success has been achieved in asymmetric fluoroalkylation reactions via nucleophilic or electrophilic processes, the development of asymmetric radical versions of this type of reactions remains a formidable challenge because of the involvement of highly reactive radical species. Here we report a catalytic asymmetric radical aminoperfluoroalkylation and aminodifluoromethylation of alkenes with commercially available fluoroalkylsulfonyl chlorides as the radical sources, providing a versatile platform to access four types of enantioenriched α-tertiary pyrrolidines bearing β-perfluorobutanyl, trifluoromethyl, difluoroacetyl and even difluoromethyl groups in excellent yields and with excellent enantioselectivity. The key to success is not only the introduction of the CuBr/chiral phosphoric acid dual-catalytic system but also the use of silver carbonate to suppress strong background and side hydroamination reactions caused by a stoichiometric amount of the in situ generated HCl. Broad substrate scope, excellent functional group tolerance and versatile functionalization of the products make this approach very practical and attractive.

[1] Department of Chemistry, South University of Science and Technology of China, Shenzhen 518055, China. * These authors contributed equally to this work. Correspondence and requests for materials should be addressed to X.-Y.L. (email: liuxy3@sustc.edu.cn).

Chiral fluorinated amines bearing fluoroalkyl groups, such as perfluoroalkyl, trifluoromethyl and difluoromethyl group, have been gaining increasing interest among medicinal chemists as important synthetic building blocks in the design of pharmaceuticals and agrochemicals because these moieties can favourably affect the physical and biological properties of compounds[1–3]. In particular, chiral amines containing a difluoromethyl group (CF$_2$H), which could act as lipophilic hydrogen bond donors and as bio-isosteres of alcohols and thiols[4–6], should be of great importance for medicinal chemistry. Thus, the synthesis of optically pure amines containing various fluoroalkyl groups has long been recognized as a preeminent goal for organic synthesis[7–9]. Although some progress has been achieved to access β-fluoroalkyl amines, these approaches often require prochiral substrates bearing pre-installed fluoroalkyl groups or stoichiometric chiral auxiliary-based strategy; thus, needing tedious multistep transformations from commerically available materials and rendering these methods less synthetically appealing[10–16]. Compared with the popularity for the preparation of α-fluoroalkyl amines via various state-of-the-art strategies[7–9], the broadly efficient and general catalytic protocols for the construction of various types of enantioenriched β-fluoroalkyl amines, using a direct fluoroalkylation strategy from readily available starting materials and reagents in an asymmetric manner, are much less developed[10–16].

In recent years, direct incorporation of fluoroalkyl groups using different types of fluoroalkylating agents in asymmetric catalytic ways has been established as a powerful technique for the sustainable preparation of chiral fluorinated molecules[7–9]. In contrast to the great success in developing nucleophilic and electrophilic fluoroalkylation reactions, the corresponding asymmetric radical fluoroalkylation versions remain scarce[17–19], largely because of the intrinsic reactivity of the involved odd-electron species[20]. On the other hand, intermolecular addition of fluoroalkyl radicals to unactivated alkenes has emerged as one of the most attractive strategies for the direct 1,2-difunctionalization of alkenes to simultaneously construct two vicinal chemical bonds by using a variety of radical fluoroalkyl precursors in racemic form[21–27]. The development of asymmetric catalytic versions of such transformations, however, still remains a formidable challenge with few successful examples. Buchwald succeeded in utilizing copper/bis(oxazoline) catalysts for asymmetric intramolecular oxytrifluoromethylation of alkenes with carboxylic acids with Togni's reagent, giving rise to good enantio-selectivities (74–83% ee)[28,29]. More recently, we discovered that a dual-catalytic system[30–33] of Cu(I) and chiral phosphoric acid (CPA) could catalyse the asymmetric radical aminotrifluoro-methylation of alkenes with Togni's reagent as the CF$_3$ radical

source with excellent enantioselectivity[34]. Given these facts, it is still very desirable to design and develop new effective catalytic systems for efficient and general particularly challenging asymmetric radical fluoroalkylation reactions with versatile fluoroalkylating reagents.

Within the burgeoning field of radical fluoroalkyl reagents[21–27], great progress has been made in the direct 1,2-difunctionalization-type fluoroalkylation of alkenes by employing stable fluoroalkylsulfonyl chlorides as the radical sources to in situ generate the desired fluoroalkyl radicals[35–39]. For the efficient collection of fundamental yet synthetically formidable chiral β-fluoroalkyl amine-building blocks directly from readily available materials and particularly intrigued by our recent success in asymmetric radical aminotrifluoromethylation of alkenes using a dual-catalytic system of Cu(I) and CPA[34], we envisaged the possibility of realizing an unprecedented asymmetric radical aminoperfluoroalkylation and aminodifluoromethylation of alkenes with various fluoroalkylsulfonyl chlorides through such a dual-catalytic system (Fig. 1). Several challenges are associated with the development of this reaction, such as (1) stoichiometric amount of strong achiral Brønsted acid HCl would be generated, which might not only result in strong background reactions against the chiral acid-catalysed process but also lead to competitive hydroamination of alkene as previous report[40]; (2) it is not easy to search for a uniform catalytic system broadly applicable to a variety of electronically distinct fluoroalkylsulfonyl chlorides, such as perfluoroalkyl, trifluoromethyl, difluoroacetyl and even difluoromethyl radical precursors. Most notable is that difluoromethylsulfonyl chloride is only used as the suitable radical source for the intramolecular aminofluoroalkylation reaction in racemic form under photoredox catalysis[37]. Herein we describe our efforts toward the development of the dual Cu(I)/CPA-catalysed asymmetric radical intramolecular aminofluoroalkylation of alkenes with various fluoroalkylsulfonyl chlorides. This mild protocol represents the general and broadly applicable platform enabling efficient access to four types of enantioenriched functionalized α-tertiary pyrrolidines bearing various β-fluoroalkyl groups with excellent yields and enantioselectivity (Fig. 1).

## Results

**Asymmetric radical aminoperfluoroalkylation of alkenes.** To probe the feasibility of our proposed assumption, we started our investigation by reacting N-alkenyl urea **1a** with perfluorobutanyl sulfonyl chloride **2a** as the model reaction. In initial studies, however, treatment of **1a** and **2a** with the previously established asymmetric aminotrifluoromethylation conditions[34] provided the desired product **3A** in a poor yield with essentially no

**Figure 1 | Our proposal.** Dual Cu(I)/CPA-catalysed asymmetric radical aminofluoroalkylation of alkenes.

**Table 1 | Screening of reaction conditions.**

(S)-**A1**: Ar = 4-Ph-C$_6$H$_4$
(S)-**A2**: Ar = 4-Cl-C$_6$H$_4$
(S)-**A3**: Ar = 1-Naphthyl
(S)-**A4**: Ar = 9-Phenanthryl
(S)-**A5**: Ar = 9-Anthryl

(S)-**A6**: Ar = 4-Ph-C$_6$H$_4$
(S)-**A7**: Ar = 1-Naphthyl

(R)-**A8**: Ar = 4-CF$_3$-C$_6$H$_4$
(R)-**A9**: Ar = 2-Naphthyl

| Entry | [Cu] | Base (x equiv.) | CPA | Solvent | Yield (%) | | ee (%)[*] |
|---|---|---|---|---|---|---|---|
| | | | | | 3A[†] | 3AA[†] | |
| 1 | CuBr | — | (S)-**A1** | EtOAc | 21 | 56 | 0 |
| 2 | CuBr | NaHCO$_3$ (1.2) | (S)-**A1** | EtOAc | 91 | 0 | 82 |
| 3 | CuBr | K$_2$CO$_3$ (0.6) | (S)-**A1** | EtOAc | 10 | Trace | 0 |
| 4 | CuBr | AgOAc (1.2) | (S)-**A1** | EtOAc | 95 | 0 | 98 |
| 5 | CuBr | Ag$_3$PO$_4$ (0.4) | (S)-**A1** | EtOAc | 14 | 0 | 33 |
| 6 | **CuBr** | **Ag$_2$CO$_3$ (0.6)** | **(S)-A1** | **EtOAc** | **98** | **0** | **90** |
| 7 | CuBr | AgOTs (1.2) | (S)-**A1** | EtOAc | 98 | 0 | 1 |
| 8 | CuBr | Ag$_2$CO$_3$ (0.6) | (S)-**A2** | EtOAc | 94 | 0 | 48 |
| 9 | CuBr | Ag$_2$CO$_3$ (0.6) | (S)-**A3** | EtOAc | 96 | 0 | 25 |
| 10 | CuBr | Ag$_2$CO$_3$ (0.6) | (S)-**A4** | EtOAc | 95 | 0 | 12 |
| 11 | CuBr | Ag$_2$CO$_3$ (0.6) | (S)-**A5** | EtOAc | 93 | 0 | 7 |
| 12 | CuBr | Ag$_2$CO$_3$ (0.6) | (S)-**A6** | EtOAc | 96 | 0 | − 75 |
| 13 | CuBr | Ag$_2$CO$_3$ (0.6) | (S)-**A7** | EtOAc | 88 | 0 | − 27 |
| 14 | CuBr | Ag$_2$CO$_3$ (0.6) | (R)-**A8** | EtOAc | 88 | 0 | 61 |
| 15 | CuBr | Ag$_2$CO$_3$ (0.6) | (R)-**A9** | EtOAc | 97 | 0 | 73 |
| 16 | CuCl | Ag$_2$CO$_3$ (0.6) | (S)-**A1** | EtOAc | 98 | 0 | 86 |
| 17 | CuI | Ag$_2$CO$_3$ (0.6) | (S)-**A1** | EtOAc | 97 | 0 | 89 |
| 18 | CuOAc | Ag$_2$CO$_3$ (0.6) | (S)-**A1** | EtOAc | 94 | 0 | 50 |
| 19 | CuBr | Ag$_2$CO$_3$ (0.6) | (S)-**A1** | DCM | 95 | 0 | 91 |
| 20 | CuBr | Ag$_2$CO$_3$ (0.6) | (S)-**A1** | THF | 50 | 0 | 13 |
| 21 | CuBr | Ag$_2$CO$_3$ (0.6) | (S)-**A1** | n-hexane | 15 | 0 | 95 |
| 22 | CuBr | Ag$_2$CO$_3$ (0.6) | (S)-**A1** | i-PrOAc | 96 | 0 | 93 |
| 23 | **CuBr** | **Ag$_2$CO$_3$ (0.6)** | **(S)-A1** | **i-PrCO$_2$Et** | **98** | **0** | **99** |
| 24[‡] | CuBr | Ag$_2$CO$_3$ (0.6) | (S)-**A1** | i-PrCO$_2$Et | 90 | 0 | 95 |
| 25 | — | Ag$_2$CO$_3$ (0.6) | (S)-**A1** | i-PrCO$_2$Et | 0 | 0 | 0 |

CPA, chiral phosphoric acid; HPLC, high-performance liquid chromatography; NMR, nuclear magnetic resonance.
Reaction conditions: **1a** (0.05 mmol), **2a** (0.06 mmol), CuBr (10 mol%), Ag$_2$CO$_3$ (0.03 mmol), CPA (5 mol%) and solvent (0.5 ml) at 28 °C for 36 h under argon.
[*]Ee value on HPLC.
[†]Yield based on $^1$H NMR analysis of the crude product using CH$_2$Br$_2$ as an internal standard.
[‡](S)-**A1** (2.5 mol%) was used.
The bold entry 6 in Table 1 shows that the best base and the chiral phosphoric acid (CPA) were found. The bold entry 23 in Table 1 indicates that the best reaction condition was confirmed.

enantiocontrol, along with side hydroamination product **3AA** in 56% yield (Table 1, entry 1). This might be attributed to the *in situ* generated stoichiometric amount of HCl, which may promote both the background and side reactions, which is in agreement with our initial assumption. Given this, we surmised that the use of a stoichiometric weak basic inorganic salt would neutralize the equivalent of strong acid HCl to generate not only the relatively weaker achiral acid such as carbonic acid but also insoluble metal chloride in organic solvents; therefore, establishing a phase separation between the

catalytic system (bulk solution) and the stoichiometric metal salt (solid phase)[30–33,41,42] to allow for the radical aminofluoroalkylation of alkenes in an enantioselective manner. Subsequently, we chose 1.2 equiv. of NaHCO$_3$ as a base to neutralize the equivalent of HCl generated during the reaction based on our initial hypothesis, and we were encouraged to observe a significant increase in product yield (up to 91%) and with good enantioselectivity (82% ee) and found that no hydroamination byproduct was observed (entry 2). A thorough evaluation of different inorganic salts indicated that they

**Table 2 | Substrate scope for aminoperfluoroalkylation of 1\*.**

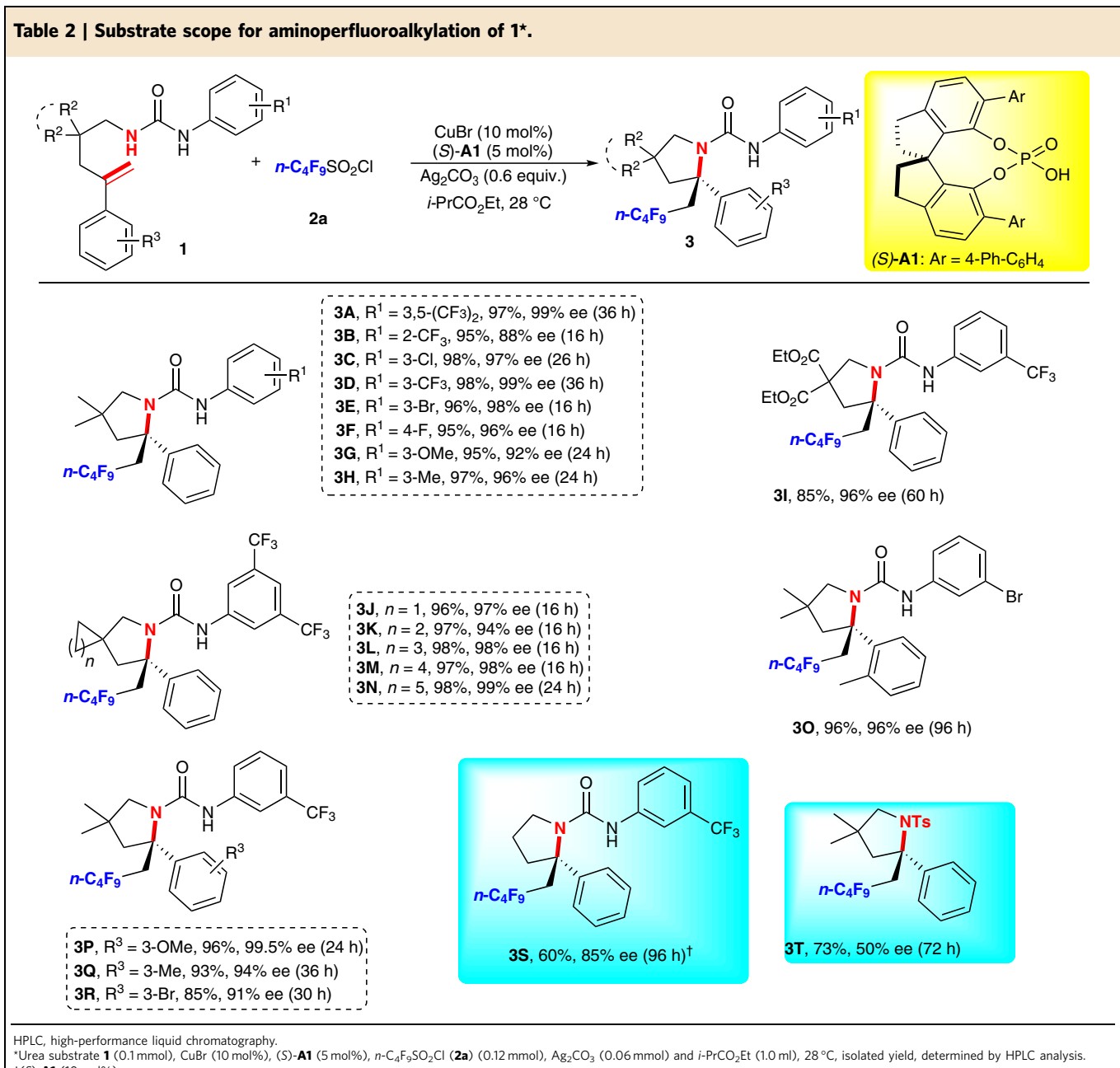

**3A**, R$^1$ = 3,5-(CF$_3$)$_2$, 97%, 99% ee (36 h)
**3B**, R$^1$ = 2-CF$_3$, 95%, 88% ee (16 h)
**3C**, R$^1$ = 3-Cl, 98%, 97% ee (26 h)
**3D**, R$^1$ = 3-CF$_3$, 98%, 99% ee (36 h)
**3E**, R$^1$ = 3-Br, 96%, 98% ee (16 h)
**3F**, R$^1$ = 4-F, 95%, 96% ee (16 h)
**3G**, R$^1$ = 3-OMe, 95%, 92% ee (24 h)
**3H**, R$^1$ = 3-Me, 97%, 96% ee (24 h)

**3I**, 85%, 96% ee (60 h)

**3J**, n = 1, 96%, 97% ee (16 h)
**3K**, n = 2, 97%, 94% ee (16 h)
**3L**, n = 3, 98%, 98% ee (16 h)
**3M**, n = 4, 97%, 98% ee (16 h)
**3N**, n = 5, 98%, 99% ee (24 h)

**3O**, 96%, 96% ee (96 h)

**3P**, R$^3$ = 3-OMe, 96%, 99.5% ee (24 h)
**3Q**, R$^3$ = 3-Me, 93%, 94% ee (36 h)
**3R**, R$^3$ = 3-Br, 85%, 91% ee (30 h)

**3S**, 60%, 85% ee (96 h)†

**3T**, 73%, 50% ee (72 h)

HPLC, high-performance liquid chromatography.
*Urea substrate **1** (0.1 mmol), CuBr (10 mol%), (S)-**A1** (5 mol%), n-C$_4$F$_9$SO$_2$Cl (**2a**) (0.12 mmol), Ag$_2$CO$_3$ (0.06 mmol) and i-PrCO$_2$Et (1.0 ml), 28 °C, isolated yield, determined by HPLC analysis.
†(S)-**A1** (10 mol%).

have significant impact on the efficiency and enantioselectivity; and Ag$_2$CO$_3$ was found to be particularly effective (entries 2–7)[37]. Noteworthy is that Ag$_2$CO$_3$ was used to improve the product yield of the intramolecular aminofluoroalkylation reaction in racemic form under photoredox catalysis[37]. Its good performance compared with other inorganic bases might be due to the generation of the highly insoluble AgCl and weaker carbonic acid to ensure unselective background reactivity is minimal. Conversely, and as expected, addition of silver compounds such as AgOTs led to significantly lower ee due to competition from the racemic reaction catalysed by in situ generated strong acid (TsOH; entry 7). We next screened different combinations of various BINOL (1,1′-bi-2-naphthol)- and SPINOL (2,2′,3,3′-tetrahydro-1,1′-spirobi[indene]-7,7′-diol) CPAs[43–47] and Cu salts (entries 6 and 8–18) and found that the dual catalyst composed of CuBr (10 mol%) and (S)-**1A** (5 mol%) with 4-Ph-phenyl group at the 3,3′-positions was the best in terms of enantioselectivity (90%

ee; entry 6). Among the solvents screened (entries 19–23), ethyl isobutyrate was found to be the most efficient one (98% yield and 99% ee; entry 23). It should be noted that the catalyst loading could be reduced from 5 to 2.5 mol% without remarkably affecting the reaction efficiency and stereoselectivity (entry 24). Only a trace amount of the desired product was obtained in the absence of a copper(I) catalyst (entry 25), revealing that copper(I) is essential as a single-electron catalyst to activate n-C$_4$F$_9$SO$_2$Cl to generate n-C$_4$F$_9$ radical.

With the optimal reaction conditions being established, we next investigated the substrate scope of the asymmetric radical aminoperfluoroalkylation and the results are summarized in Table 2. First, a series of substrates with N-aryl urea groups were investigated. The results revealed that both the position and electronic property of the substituents on the aromatic ring (R$^1$) have a negligible effect on the reaction efficiency and stereoselectivity of this radical reaction. For example, a range of

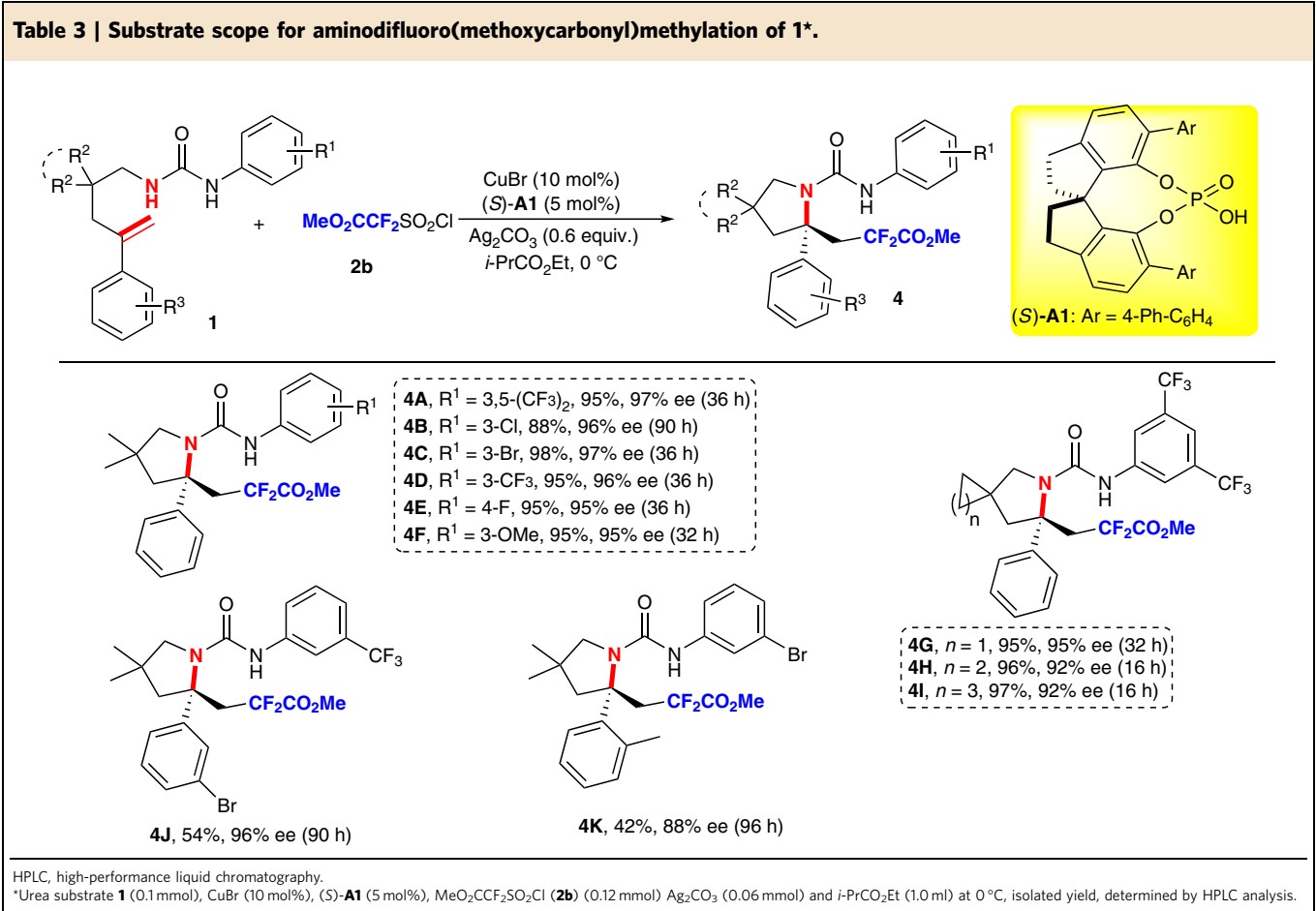

**Table 3 | Substrate scope for aminodifluoro(methoxycarbonyl)methylation of 1\*.**

HPLC, high-performance liquid chromatography.
\*Urea substrate **1** (0.1 mmol), CuBr (10 mol%), (*S*)-**A1** (5 mol%), MeO₂CCF₂SO₂Cl (**2b**) (0.12 mmol) Ag₂CO₃ (0.06 mmol) and *i*-PrCO₂Et (1.0 ml) at 0 °C, isolated yield, determined by HPLC analysis.

diversely functionalized alkenyl ureas **1**, including those having aryl groups with electron-withdrawing (CF₃, F, Cl and Br) or electron-donating groups (OMe and Me) at different positions (*ortho*, *meta* or *para*) on the phenyl ring, as well as bistrifluoromethyl substituents, were found to be suitable substrates to afford the expected products **3A–3H** in 95–98% yields with 88–99% ee. The absolute configuration of (*R*)-**3D** was determined by X-ray crystallographic analysis (Supplementary Fig. 1), and those of other perfluoroalkyl-containing products were determined in reference to **3D**. Next, under the almost identical reaction conditions, substrate **1i** proceeded well to give **3I** in 85% yield and 96% ee, revealing that the reaction was not significantly influenced by changing the tether group to a di-ester group. To our delight, substrates containing three- to seven-membered rings were all suitable for the reaction to produce the enantioenriched perfluoroalkylated spiro products **3J–3N** in excellent yields with 94–99% ee. In addition, other *gem*-disubstituted alkenes bearing different substituents (R³) on phenyl ring were tested as well. β-Perfluoroalkyl amines **3O–3R** were obtained in 85–96% yields with excellent ee (91 to >99% ee), demonstrating the excellent tolerance of substituents bearing different electronic and steric nature. Unlike tethered substrates, the use of nonbranched substrates without the Thorpe–Ingold effects is generally far less studied in previous related process, probably due to the unfavourable entropy factor and proximity effects in the cyclic transition state of such processes[48]. It is more encouraging to note that the unbranched substrate **1s** underwent the current aminoperfluoroalkylation reaction smoothly to deliver the desired product **3S** in 60% yield with 85% ee under the same conditions, which was in sharp

contrast to asymmetric radical aminotrifluoromethylation with Togni's reagent as the CF₃ source to give low product yield and ee in case of such a substrate[34]. Thus, the scope of this new reaction is substantially expanded. Even more remarkably, other protected amines could also be employed in the reaction. Under the conditions identical to those of aminoperfluoroalkylation of alkenyl ureas, the reaction of *N*-alkenyl tosyl **1t** also afforded the corresponding product **3T** in 73% yield, albeit with 50% ee, which is currently under further optimization in our laboratory. Unfortunately, when substrates **1U** without the aromatic substituent and **1V** bearing the benzylic substituent were used as the substrate, a trace amount of the desired cyclization product was observed along with the chlorine addition products **3U** and **3V** in 86% and 85% yield, respectively (Supplementary Fig. 2).

**Asymmetric aminodifluoro(methoxycarbonyl)methylation**. Due to its membrane permeability, difluoromethyl group (CF₂R) is routinely employed in search for lead structures in drug discovery in recent years[16], and has thus inspired the development of new reagents and strategies for the synthesis of difluoromethyl-containing compounds[49–58]. However, there are very few catalytic asymmetric approaches to access such chiral building blocks containing difluoromethyl groups, using a direct fluoroalkylation strategy[58]. We were naturally eager to extend the methodology to other fluoroalkylsulfonyl chlorides as radical precursors for the synthesis of the potentially useful chiral β-difluoromethyl or difluoroacetyl amines. To our delight, the reaction of *N*-alkenyl urea substrate **1a** with **2b** under the otherwise identical reaction conditions delivered difluoroacetyl-containing product **4A** in

**Table 4 | Substrate scope for aminodifluoromethylation of 1\*.**

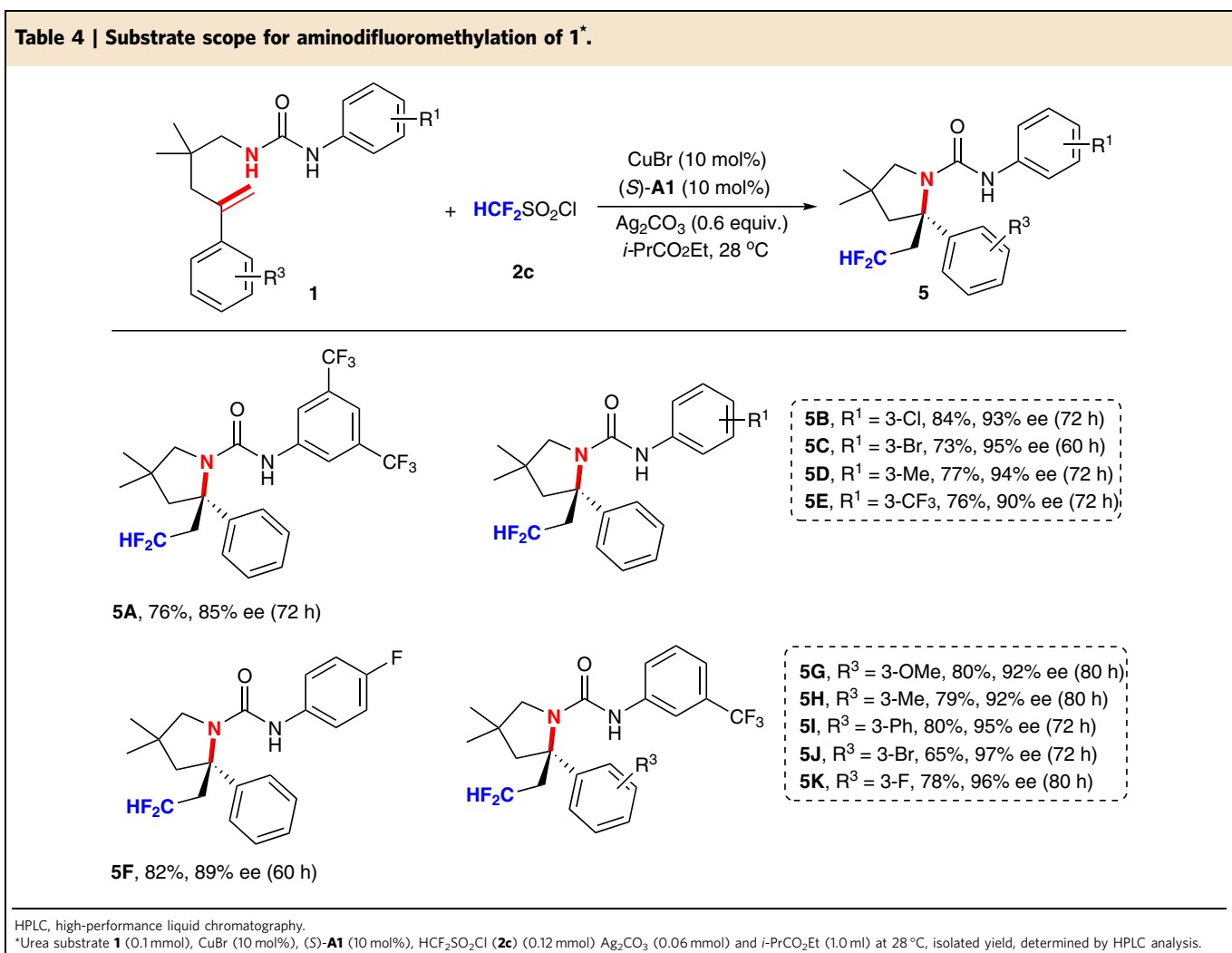

HPLC, high-performance liquid chromatography.
\*Urea substrate **1** (0.1 mmol), CuBr (10 mol%), (S)-**A1** (10 mol%), HCF₂SO₂Cl (**2c**) (0.12 mmol) Ag₂CO₃ (0.06 mmol) and i-PrCO₂Et (1.0 ml) at 28 °C, isolated yield, determined by HPLC analysis.

97% yield with 85% ee, which containing an acetyl structure is especially appealing because further transformations of the acetyl group in the obtained product would give access to a more diverse class of structures[59–60]. Further improvement of the stereoselectivity indicated that lowering the reaction temperature was obviously beneficial for such a process, giving **4A** in 95% yield with 97% ee at 0 °C (Table 3). N-Aryl urea substrates with various substituents (R¹) on the phenyl ring were also explored. It was found that those bearing electron-withdrawing (F, Cl, Br and CF₃) or electron-donating groups (OMe) at different positions (*para* or *meta*) on the aromatic ring all reacted smoothly to afford the corresponding products **4B–4F** in excellent yields with 95–97% ee. Moreover, the expected spiro products **4G–4I** were also obtained in excellent yields with 92–95% ee. Similarly, other substrates with substituents (R³) were also applicable under the standard conditions, affording products **4J** and **4K** in moderate yields with 96% and 88% ee, respectively.

**Asymmetric radical aminodifluoromethylation of alkenes.** Inspired by the above success, we thus switched our synthetic targets to collect chiral β-difluoromethyl amines (Table 4). As expected, the reaction of N-alkenyl urea substrate **1a** with difluoromethylsulfonyl chloride **2c** under the almost identical reaction conditions furnished the desired β-difluoromethyl amine **5A** in 76% yield with 85% ee. Further exploration of the substrate scope exemplified that mono-substituents on *meta* or *para*

position of N-aryl ring offered products **5B–5F** with a higher ee (89–95% ee) than **5A**. Similarly, the substituent (R³) on the *meta* position of phenyl ring was also investigated and the observed results indicated that the stereoselectivity was not significantly influenced by the electronic property to give β-difluoromethyl amines **5G–5K** with excellent enantioselectivity (92–97% ee).

**Asymmetric radical aminotrifluoromethylation of alkenes.** We then expanded the scope to the catalytic asymmetric radical aminotrifluoromethylation of alkenes[34] in the presence of trifluoromethanesulfonyl chloride (CF₃SO₂Cl; **2d**) as the radical CF₃ source, considering the fact that, as compared with Togni's reagent, **2d** is a bench stable and inexpensive radical CF₃ reagent and produces SO₂ and inorganic chloride as the by-products that could be removed effectively from the reaction mixture during the work-up. To our delight, under the otherwise identical reaction conditions, the reaction of N-alkenyl urea substrate **1a** with **2d** afforded trifluoromethyl-containing product **6A** in 91% yield with 93% ee in the presence of 5 mol% of chiral catalyst (S)-**A1**, which showed markedly enhanced reactivity as compared with that using Togni's reagent as the CF₃ source (Table 5). For other N-alkenyl urea substrates, the reaction also proceeded smoothly, even at 0 °C, to afford the corresponding products **6B–6D** in 96–98% yields with 90–96% ee. Meanwhile, it is striking to note that substrates with substituents R³ on the aromatic ring

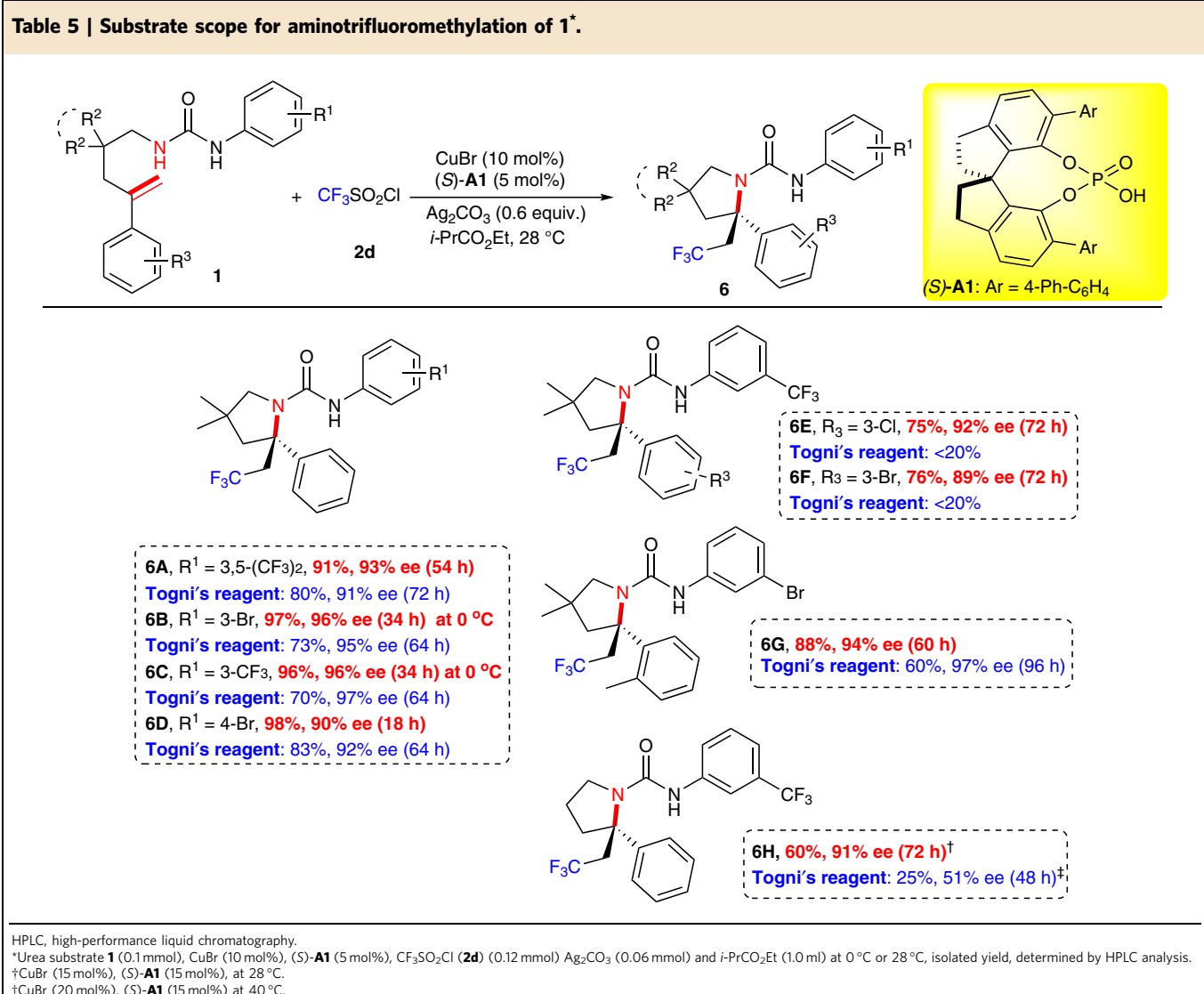

**Table 5 | Substrate scope for aminotrifluoromethylation of 1\*.**

HPLC, high-performance liquid chromatography.
\*Urea substrate **1** (0.1 mmol), CuBr (10 mol%), (S)-**A1** (5 mol%), CF₃SO₂Cl (**2d**) (0.12 mmol) Ag₂CO₃ (0.06 mmol) and i-PrCO₂Et (1.0 ml) at 0 °C or 28 °C, isolated yield, determined by HPLC analysis.
†CuBr (15 mol%), (S)-**A1** (15 mol%), at 28 °C.
‡CuBr (20 mol%), (S)-**A1** (15 mol%) at 40 °C.

were also applicable, giving products **6E–6G** in good to excellent yields with 89–94% ee, most of which were difficult to access using our previous method[34]. It is more encouraging to note that the unbranched substrate **1s** underwent the current aminotrifluoromethylation reaction smoothly to deliver the desired product **6H** in 60% yield with 91% ee, which showed dramatically enhanced reactivity as compared with that using Togni's reagent as the CF₃ source to give low product yield and ee in case of such a substrate[34]. These results indicated that trifluoromethanesulfonyl chloride as the CF₃ source exhibited a number of clear advantages in terms of high reactivity, low-cost and simple work-up over its Togni's reagent-based counterpart and rendering the current reaction system a more appealing alternative to the previous approach[34].

**Synthetic application**. An important synthetic application of the present strategy is that the obtained enantioenriched products can serve as pivotal intermediates for easy access to other medicinally intriguing fluoro-containing amines. For example, simple reduction or hydrolysis of ester group smoothly generated the corresponding fluoro-containing alcohol **7** and carboxylic acid **8** in excellent yields without diminishing the enantioselectivity (Fig. 2, equations a and b). Besides, the convenient

transformation of **4A** in the presence of [bis(trifluoroacetoxy)-iodo]benzene provided tricyclic amine **9** bearing an α-tetra-substituted carbon stereocenter in 43% yield with no loss in the enantioselectivity (equation c). Furthermore, treatment of difluoroacetyl amines **4F** with BH₃·SMe₂ afforded an unexpected difluoro-containing pyrrolidine **10** in 43% yield with the retention of enantioselectivity (equation d). It should be emphasized that the pyrrolidine is a central structural motif in a variety of natural alkaloids and pharmaceutical compounds[61].

**Mechanistic investigations**. To gain some insights into the reaction mechanism, a series of control experiments were conducted. First, a radical-trapping experiment using 2,2,6,6-tetramethyl-1-piperidinyloxy as a radical scavenger was carried out and it is found that the present reaction was completely shut down (Fig. 3, equation 1). Next, when a standard radical clock cyclopropane moiety was treated with perfluorobutanyl sulfonyl chloride **2a** under the standard conditions, the expected amino-perfluoroalkylation product **12** was hardly observed. Instead, n-C₄F₉-containing product **13** as a mixture of E/Z isomers was obtained in 85% yield via a radical addition/cyclopropane ring opening/chloride-trapping cascade process (Fig. 3, equation 2). These observations, together with previous studies on the radical

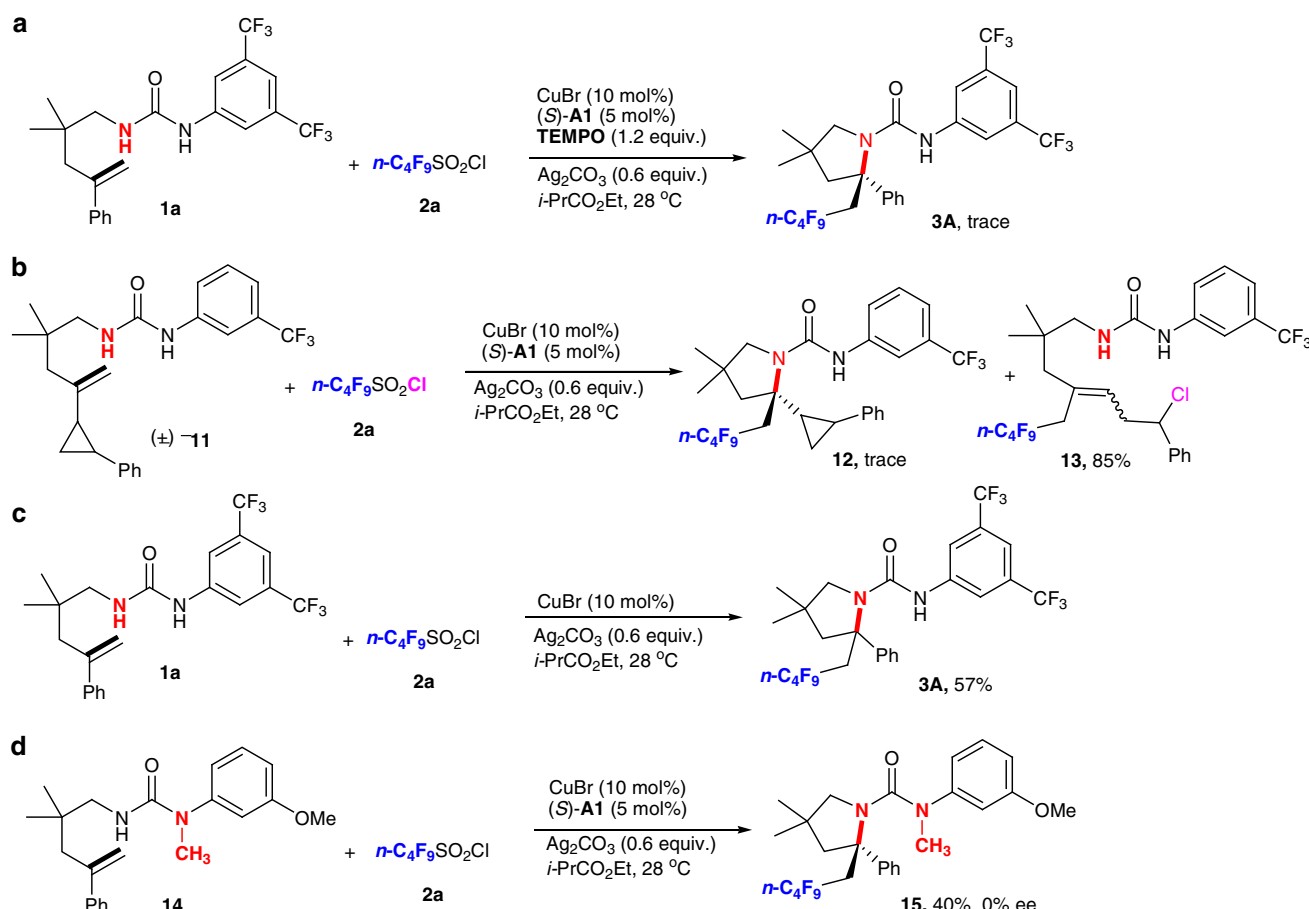

**Figure 2 | Versatile transformations.** (**a,b**) Reduction and hydrolysis reaction. (**c**) Cyclization reaction. (**d**) Reduction to a difluoro-containing pyrrolizidine. KHMDS, Potassium bis(trimethylsilyl)amide; PIFA, [Bis(trifluoroacetoxy)iodo]benzene.

**Figure 3 | Mechanistic study.** (**a**) Trapping with TEMPO. (**b**) Radical clock. (**c,d**) Control reactions. TEMPO, 2,2,6,6-tetramethyl-1-piperidinyloxy.

aminodifluoromethylation of alkene with **2a** by Cu(I) catalyst[37], suggest that the $n$-C$_4$F$_9$ radical is likely involved as the reactive species for further addition of alkene to generate α-R$_f$ alkyl radical **C** under the current reaction conditions. To further confirm the dual roles of CuBr and CPA, treatment of **1a** with **2a** under otherwise identical conditions in the presence of either CuBr alone or (*S*)-**A1** alone (see Fig. 3, equation 3, and Table 1, entry 25) gave the corresponding product **3A** in lower yields; thus, revealing that both the Cu(I) salt and the phosphoric acid are necessary for this reaction, and the phosphoric acid could play an important role in the activation of perfluorobutanyl sulfonyl chloride. In contrast, the control reaction of methyl-protected urea derivative **14** with **2a** under the standard conditions furnished the desired product **15** in only 40% yield with no

**Figure 4 | Mechanistic proposal.** Two pathways were tentatively proposed.

enantioselectivity, clearly indicating that the urea with two acidic N–H at the appropriate position is critical for its activation to improve reaction efficiency and control asymmetric induction.

On the basis of above mechanistic investigations and previous studies[28,29,34,37,62], a working mechanism for the dual Cu(I)/CPA-catalysed asymmetric radical aminoperfluoroalkylation and aminodifluoromethylation is tentatively proposed in Fig. 4. Initially, the $R_f$ radical and chiral monophosphate or bisphosphate Cu(II) **B** or **B'** are generated from the single-electron transfer reaction of the corresponding $R_fSO_2Cl$ with CuBr and the phosphoric acid, along with the generation of a stoichiometric amount of sulfur dioxide and chloride anion. The chiral bisphosphate Cu(II) **B'** is possibly formed via halide abstraction from the monophosphate Cu(II) **B** with $Ag_2CO_3$ to help tight association of Cu with chiral couteranion[30]. Here $Ag_2CO_3$ acts as a chloride scavenger via the formation of insoluble AgCl in organic solution, thereby minimizing the strong acid HCl-associated unselective background reactions. The subsequent addition of $R_f$ radical to alkene gives the α-$R_f$ alkyl radical **C**, which could be trapped by Cu(II) phosphate **B** or **B'** to form a Cu(II) species **D**, in which alkyl radical intermediate could be trapped by Cu(II) phosphate to generate a Cu(III) species **E** (refs 28,29,34,63–68; path a). During this process, the chiral phosphate could control the facial selectivity of such reaction via both hydrogen-bonding interactions with the N–H bond adjacent to the aryl group and ion-pairing interactions in a concerted transition state. Then, reductive elimination of the resulting Cu(III) species **E** would afford the final product **3** along with the regeneration of the copper Cu(I) and the phosphoric acid. However, another pathway (path b) via single-electron oxidization of intermediate **D** to the corresponding carbocation intermediate **F**, which undergoes C–N bond formation to give final product **3**, could not be excluded at the present stage. Therefore, rigorous investigations are necessary to unambiguously elucidate the exact mechanism.

## Discussion
We have achieved the first catalytic asymmetric radical aminoperfluoroalkylation and aminodifluoromethylation of alkenes with commercially available fluoroalkylsulfonyl chlorides. Critical to the success of this process is not only the introduction of the CuBr/CPA dual-catalytic system but also the use of silver

carbonate to suppress background and side hydroamination reactions caused by a stoichiometric amount of the *in situ* generated HCl. This approach offers a sustainable and broadly applicable platform enabling efficient access to four types of enantioenriched functionalized α-tertiary pyrrolidines bearing versatile β-fluoroalkyl groups with excellent efficiency, remarkable enantioselectivity and excellent functional group tolerance. Noteworthy is that the newly developed asymmetric aminotrifluoromethylation of alkenes with trifluoromethanesulfonyl chloride as the $CF_3$ source has obvious advantages in terms of high reactivity, low-cost and simple work-up as compared with that using Togni's reagent as the $CF_3$ source, rendering the method to be a valuable alternative to the previous approach[34]. Furthermore, this transformation enables the efficient construction of other useful chiral fluoroalkyl-containing building blocks. Further studies including the expansion to more radical precursors and the development of a more challenging intermolecular catalytic asymmetric version are ongoing in our laboratory.

## Methods
**Asymmetric radical aminoperfluoroalkylation of alkenes.** Under argon, an oven-dried resealable Schlenk tube equipped with a magnetic stir bar was charged with urea substrate **1** (0.1 mmol, 1.0 equiv.), CuBr (1.43 mg, 0.01 mmol, 10 mol%), CPA (S)-**A1** (3.1 mg, 0.005 mmol, 5 mol%), $Ag_2CO_3$ (16.56 mg, 0.06 mmol, 0.6 equiv.), n-$C_4F_9SO_2Cl$ (**2a**) (38.15 mg, 0.12 mmol, 1.2 equiv.) and ethyl isobutyrate (1.0 ml) at 28 °C, and the sealed tube was then stirred at 28 °C. Upon completion (monitored by thin-layer chromatography (TLC)), the reaction mixture was directly purified by a silica gel chromatography (eluent: petroleum ether/EtOAc = 100/0–5/1, using petroleum ether (100%) to remove the solvent (ethyl isobutyrate) at first) to afford the desired product **3**.

**Asymmetric aminodifluoro(methoxycarbonyl)methylation.** Under argon, an oven-dried resealable Schlenk tube equipped with a magnetic stir bar was charged with urea substrate **1** (0.1 mmol, 1.0 equiv.), CuBr (1.43 mg, 0.01 mmol, 10 mol%), CPA (S)-**A1** (3.1 mg, 0.005 mmol, 5 mol%), $Ag_2CO_3$ (16.56 mg, 0.06 mmol, 0.6 equiv.), $MeO_2CCF_2SO_2Cl$ (**2b**) (25 mg, 0.12 mmol, 1.2 equiv.) and ethyl isobutyrate (1.0 ml) at 0 °C, and the sealed tube was then stirred at 0 °C. Upon completion (monitored by TLC), the reaction mixture was directly purified by a silica gel chromatography (eluent: petroleum ether/EtOAc = 100/0–5/1, using petroleum ether (100%) to remove the solvent (ethyl isobutyrate) at first) to afford the desired product **4**.

**Asymmetric radical aminodifluoromethylation of alkenes.** Under argon, an oven-dried resealable Schlenk tube equipped with a magnetic stir bar was charged with urea substrate **1** (0.1 mmol, 1.0 equiv.), CuBr (1.43 mg, 0.01 mmol, 10 mol%),

CPA (S)-**A1** (6.2 mg, 0.01 mmol, 10 mol%), Ag$_2$CO$_3$ (16.56 mg, 0.06 mmol, 0.6 equiv.), HCF$_2$SO$_2$Cl (**2c**) (18.0 mg, 0.12 mmol, 1.2 equiv.) and ethyl isobutyrate (1.0 ml) at 28 °C, and the sealed tube was then stirred at 28 °C. Upon completion (monitored by TLC), the reaction mixture was directly purified by a silica gel chromatography (eluent: petroleum ether/EtOAc = 100/0–5/1, using petroleum ether (100%) to remove the solvent (ethyl isobutyrate) at first) to afford the desired product **5**.

**Asymmetric radical aminotrifluoromethylation of alkenes.** Under argon, an oven-dried resealable Schlenk tube equipped with a magnetic stir bar was charged with urea substrate **1** (0.1 mmol, 1.0 equiv.), CuBr (1.43 mg, 0.01 mmol, 10 mol%), CPA (S)-**A1** (3.1 mg, 0.005 mmol, 5 mol%), Ag$_2$CO$_3$ (16.56 mg, 0.06 mmol, 0.6 equiv.), CF$_3$SO$_2$Cl (**2d**) (20.16 mg, 0.12 mmol, 1.2 equiv.) and ethyl isobutyrate (1.0 ml) at 0 °C or 28 °C, and the sealed tube was then stirred at 0 or 28 °C. Upon completion (monitored by TLC), the reaction mixture was directly purified by a silica gel chromatography (eluent: petroleum ether/EtOAc = 100/0–5/1, using petroleum ether (100%) to remove the solvent (ethyl isobutyrate) at first) to afford the desired product **6**.

For nuclear magnetic resonance and high-performance liquid chromatography spectra, see Supplementary Figs 3–214.

**Data availability.** The X-ray crystallographic coordinates for structures reported in this article have been deposited at the Cambridge Crystallographic Data Centre (CCDC), under deposition number CCDC 1505476 ((R)-**3D**). The data can be obtained free of charge from The Cambridge Crystallographic Data Centre via http://www.ccdc.cam.ac.uk/data_request/cif. Any further relevant data are available from the authors upon reasonable request.

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

## Acknowledgements

This paper is dedicated to Professor Chi-Ming Che on the occasion of his 60th birthday. This study is financially supported by the National Natural Science Foundation of China (nos 21572096, 21602098 and 21302088), Shenzhen overseas high level talents innovation plan of technical innovation project (KQCX20150331101823702), Shenzhen special funds for the development of biomedicine, Internet, new energy, and new material industries (JCYJ20150430160022517), and the National Key Basic Research Program of China (973 Program; 2013CB834802) is greatly appreciated.

## Author contributions

J.-S.L., F.-L.W. and X.-Y.D. performed the experiments. W.-W.H., S.C. and Y.Y. helped with characterizing all new compounds. X.-Y.L. conceived and directed the project and wrote the paper.

## Additional information

**Competing interests:** The authors declare no competing financial interests.

**Publisher's note**: 

