## [Peer review file · Nature Communications]

Reviewers' comments:

Reviewer #1 (Remarks to the Author):

Selective introduction of the fluorine-containing groups into organic compounds is an important issue in organic synthesis. Prof. Xin-Yuan Liu and co-authors developed a method for asymmetric radical amino-fluoroalkylation in the presence of the chiral catalyst. The authors insist on the novelty of asymmetric addition of fluoroalkyl radicals. However, they ignored the related report (by Prof. Miyabe) involving asymmetric fluoroalkyl radical-addition (JOC 2012, 77, 8588). Furthermore, all the reactions in this report are too particular and tacky; the substrate scope is narrow. Only the specialized substrates are applicable and the products are not of general interest to readers of Nature Communications. I do not find enough urgency in this report in order to be acceptable for publication in Nature Communications. Therefore, I suggest resubmission of this work to the specialized journals of chemistry.

Reviewer #2 (Remarks to the Author):

This is a high-quality paper on the enantioselective radical aminoperfluoroalkylation of alkenes. The authors have done a very good job in developing this asymmetric process, demonstrating good scope and excellent enantioselectivities. The SI is also done very well. I very much enjoyed reading this paper and recommend it for publication in Nature Comm.

Reviewer #3 (Remarks to the Author):

This manuscript describes an asymmetric aminoperfluoroalkylation of alkenes with fluoroalkylated sulfonyl chlorides by using the combination of copper catalyst and chiral phosphoric acid. Although the catalytic system was almost the same as the previous asymmetric aminotrifluoromethylation with Togni's reagent reported by the same authors (ref. 33), this method can provide highly optically active pyrrolidines with not only CF₃ but also C₄F₉, CF₂H, and CF₂COOMe by using fluoroalkyl sulfonyl chlorides as the fluoroalkyl source. Although the concept for the asymmetric induction is same as previously reported reaction using Togni's Reagent, the variation of commercially available fluoroalkyl sulfonyl chloride is much wider and the reaction conditions reported here could be useful and versatile. Therefore this reviewer support this work to be publishable in Nature Commun after revision of the following points.

1. In page 3, line 8, the author described that "It is very challenging to search for a uniform catalytic system~"

The reviewer agrees that the enantioselective transformation described here should be important and challenging. But it sounds like a little over statement, because the catalytic system is almost the same as it reported by the same authors, and this study may be considered as its extensive study using other fluoroalkyl sources. At the same time, beneficial effect of Ag₂CO₃ for the reaction using fluorosulfonyl chloride has also been already reported in the ref. 36. Although the authors described that they found the effect of Ag₂CO₃, proper citation is desirable.

2. In page 3, line 15, the author described that "this mild protocol represents the first~".

However, the first report should be the previous asymmetric aminotrifluoromethylation with Togni's reagent reported by the author (ref. 33). Although the fluoroalkyl radical source was different, most of the substrates are the same and the basic system for the asymmetric induction is the same. In addition, "quick access" may be overestimate because the reaction requires relatively longer reaction time.

3. In page 6, line 13, the author compared the reactivity of the C₄F₉SO₂Cl toward 1s with Togni's reagent.

However, this comparison may be insignificant due to the difference of the number of fluorinated carbons. C₄F₉-variant of Togni's reagent should be used instead of it if the author would compare the reactivity of them. Alternatively, trifluoromethylation reaction of 1s using the current method should be tried, and difference between the results of experiments using Togni's reagent and trifluoromethylsulfonyl chloride should be discussed.

4. Comments on the scope and limitation of the substrate should be added. All substrate shown are the terminal aryl-alkyl-substituted alkene. What happen in the case of the alkene without the aromatic substituent? If the aromatic group is essential, it may support the single electron oxidation pathway rather than the alkyl Cu(III) intermediate mechanism the author proposed.

5. Table 3. The word "difluoroacetylation" is not correct. It means addition of "CF₂HCO". It may be "difluoro(methoxycarbonyl)methylation"? Please reconsider the name.

6. Page 9, line 8: a lower loading of the chiral catalyst (5 mol%) was emphasized, but Table 2 and 3 also use only 5 mol%. It is rather better to say "a higher loading (10 mol%) is required in case of difluoromethylation reaction".

7. Page 12, line 11: The subsequent radical addition of R_f radical to... "radical" seems to be redundant.

8. About the mechanism proposed, the Br anion stays on Cu(II)-CPA, but in the presence of Ag₂CO₃, Br anion could also be trapped by Ag. According to Table 1, CuBr, CuCl, CuI generally gave high ee, in contrast, CuOAc gave much lower ee. It may be because of the trap effect of halogen by Ag to help tight association with CPA? It is desirable to discuss on the effects of counter anion.

Our Responses to the Comments of the Referees

Referee 2

Comment 1: *This is a high-quality paper on the enantioselective radical aminoperfluoroalkylation of alkenes. The authors have done a very good job in developing this asymmetric process, demonstrating good scope and excellent enantioselectivities. The SI is also done very well. I very much enjoyed reading this paper and recommend it for publication in Nature Comm.*

Our response: We very much appreciate these comments of the referee and sincerely thank the referee for recommending publication of the work.

Referee 3

This manuscript describes an asymmetric aminoperfluoroalkylation of alkenes with fluoroalkylated sulfonyl chlorides by using the combination of copper catalyst and chiral phosphoric acid. Although the catalytic system was almost the same as the previous asymmetric aminotrifluoromethylation with Togni's reagent reported by the same authors (ref. 33), this method can provide highly optically active pyrrolidines with not only CF₃ but also C₄F₉, CF₂H, and CF₂COOMe by using fluoroalkyl sulfonyl chlorides as the fluoroalkyl source. Although the concept for the asymmetric induction is same as previously reported reaction using Togni's Reagent, the variation of commercially available fluoroalkyl sulfonyl chloride is much wider and the reaction conditions reported here could be useful and versatile. Therefore this reviewer support this work to be publishable in Nature Commun after revision of the following points.

Our response: We very much appreciate these comments of the referee and sincerely thank the referee for recommending publication of the work after revisions.

Comment 1: *In page 3, line 8, the author described that *It is very challenging to search for a uniform catalytic system~*"The reviewer agrees that the enantioselective transformation described here should be important and challenging. But it sounds like a little over statement, because the catalytic system is almost the same as it reported by the same authors, and this study may be considered as its extensive study using other fluoroalkyl sources.*

Our response: We thank the referee for pointing out this issue. In the original text, *It is very challenging to search for a uniform catalytic system~*" refers to a uniform catalytic system broadly applicable to a variety of electronically distinct fluoroalkyl sulfonyl chlorides, such as perfluoroalkyl, trifluoromethyl, difluoroacetyl-, and even difluoromethyl radical precursors. To the best of our knowledge, only one case has been reported about the intramolecular aminofluoroalkylation of alkenes using the difluoromethyl sulfonyl chloride as the suitable radical source in racemic form under photoredox catalysis, in which other electronically distinct fluoroalkyl sulfonyl chlorides, such as perfluoroalkyl, trifluoromethyl or difluoroacetyl-group are not compatible. Nonetheless, according to this valuable suggestion, *It is very challenging to search for a uniform catalytic system~*" has been changed to "*It is not easy to search for a uniform catalytic system~*" in the revised manuscript.

At the same time, beneficial effect of Ag₂CO₃ for the reaction using fluorosulfonyl chloride has also

been already reported in the ref. 36. Although the authors described that they found the effect of Ag_2CO_3 , proper citation is desirable.

Our response: We thank the referee for pointing out this issue. We have cited this reference and highlighted this work in this part. Please kindly see “A thorough evaluation of different inorganic salts indicated that they have significant impact on the efficiency and enantioselectivity; and Ag_2CO_3 was found to be particularly effective (entries 2-7)³⁷. Noteworthy is that Ag_2CO_3 was used to improve the product yield of the intramolecular aminofluoroalkylation reaction in racemic form under photoredox catalysis³⁷.” has been added in the revised manuscript.

Comment 2: *In page 3, line 15, the author described that this mild protocol represents the first~" However, the first report should be the previous asymmetric aminotrifluoromethylation with Togni's reagent reported by the author (ref. 33). Although the fluoroalkyl radical source was different, most of the substrates are the same and the basic system for the asymmetric induction is the same. In addition, "quick access" may be overestimate because the reaction requires relatively longer reaction time.*

Our response: We thank the referee for bringing this issue to our attention and completely agree on this suggestion. “This mild protocol represents the first, to the best of our knowledge, general and broadly applicable platform enabling quick and efficient access to four types of enantioenriched functionalized α -tertiary pyrrolidines bearing various β -fluoroalkyl groups with excellent yields and enantioselectivity (Fig. 1).” has been changed to “This mild protocol represents the general and broadly applicable platform enabling efficient access to four types of enantioenriched functionalized α -tertiary pyrrolidines bearing various β -fluoroalkyl groups with excellent yields and enantioselectivity (Fig. 1).”

Comment 3: *In page 6, line 13, the author compared the reactivity of the $\text{C}_4\text{F}_9\text{SO}_2\text{Cl}$ toward **1s** with Togni's reagent. However, this comparison may be insignificant due to the difference of the number of fluorinated carbons. C_4F_9 -variant of Togni's reagent should be used instead of it if the author would compare the reactivity of them. Alternatively, trifluoromethylation reaction of **1s** using the current method should be tried, and difference between the results of experiments using Togni's reagent and trifluoromethylsulfonyl chloride should be discussed.*

Our response: We really appreciate the referee for this valuable suggestion. According to this valuable comment, we have made efforts to investigate the intramolecular trifluoromethylation of the unbranched substrate **1s** using the current method. It is more encouraging to note that the unbranched substrate **1s** underwent the current aminotrifluoromethylation reaction smoothly to deliver the desired product **6H** in 60% yield with 91% ee, which showed dramatically enhanced reactivity as compared with that using Togni's reagent as the CF_3 source to give low product yield and ee in case of such a substrate (See the result as shown below). This result has been added and discussed in the revised manuscript and Supplementary Information.

DAD1 B, Sig=254,4 Ref=off (LJSIS-6-IS-6-75C RACE SEC ODH-90-10-05.D)

Signal 2: DAD1 B, Sig=254,4 Ref=off

Peak #	RetTime [min]	Type	Width [min]	Area [mAU*s]	Height [mAU]	Area %
1	14.835	BB	0.3396	2155.80127	96.88145	50.3031
2	18.343	BB	0.5438	2129.82007	58.91344	49.6969

Totals : 4285.62134 155.79489

DAD1 B, Sig=254,4 Ref=off (LJSIS-6-143B PURI ODH-90-10-03.D)

Signal 2: DAD1 B, Sig=254,4 Ref=off

Peak #	RetTime [min]	Type	Width [min]	Area [mAU*s]	Height [mAU]	Area %
1	15.038	BB	0.4191	7270.91748	264.16098	95.5101
2	19.042	MF	0.6711	341.80203	8.48827	4.4899

Totals : 7612.71951 272.64925

Comment 4: Comments on the scope and limitation of the substrate should be added. All substrate shown are the terminal aryl-alkyl-substituted alkene. What happen in the case of the

alkene without the aromatic substituent? If the aromatic group is essential, it may support the single electron oxidation pathway rather than the alkyl Cu(III) intermediate mechanism the author proposed.

Our Response: We really appreciate the referee for this valuable suggestion. According to this comment, we have made efforts to investigate the intramolecular aminoperfluoroalkylation of substrates **1U** without the aromatic substituent and **1V** with the benzylic substituent. However, a trace amount of the desired cyclization products was observed along with the chlorine addition products **3U** and **3V** in 86% and 85% yield, respectively. The similar result was also observed in the previous reported studies (*Org. Lett.* 17, 3528-3531 (2015)). These results showed that R_f radical should initially react with alkene without the aromatic substituent to form an alkyl radical which can then be oxidized by the Cu(II) to form a carbocation, which can then itself be trapped by chlorine anion (*Org. Lett.* 17, 3528-3531 (2015)). These results have already been discussed in the revised manuscript. Please kindly note that “Unfortunately, when substrates **1U** without the aromatic substituent and **1V** bearing the benzylic substituent were used as the substrate, a trace amount of the desired cyclization products was observed along with the chlorine addition products **3U** and **3V** in 86% and 85% yield, respectively (Supplementary Fig. 2).” (see the results as shown below).

However, in the present work, we think that the exact mechanism for this reaction remains unclear at the present stage and deserves further detailed studies. According to your valuable suggestions, two possible reaction pathways for the asymmetric radical aminoperfluoroalkylation and aminodifluoromethylation are proposed in the revised manuscript (see Figure 4 in the revised manuscript), although further studies are needed to clarify the details. Please kindly see the revised part about the mechanism discussion as shown below.

“On the basis of above mechanistic investigations and previous studies^{28-29,34,37,62}, a working mechanism for the dual Cu(I)/chiral phosphoric acid-catalyzed asymmetric radical aminoperfluoroalkylation and aminodifluoromethylation is tentatively proposed in Fig. 4. Initially, the R_f radical and chiral monophosphate or bisphosphate Cu(II) **B** or **B'** are generated from the single electron transfer reaction of the corresponding R_fSO₂Cl with CuBr and the phosphoric acid, along with the generation of a stoichiometric amount of sulfur dioxide and chloride anion. The chiral bisphosphate Cu(II) **B'** is possibly formed via halide abstraction from the monophosphate Cu(II) **B** with Ag₂CO₃ to help tight association of Cu with chiral counteranion³⁰. Here, Ag₂CO₃ acts as a chloride scavenger via the formation of insoluble AgCl in organic solution, thereby minimizing the strong acid HCl-associated unselective background reactions. The subsequent addition of R_f radical to alkene gives the α-R_f alkyl radical **C**, which could be trapped by Cu(II) phosphate **B** to form a Cu(II) species **D**, in which alkyl radical intermediate could be trapped by Cu(II) phosphate to generate a Cu(III) species **E**^{28-29,34,63-68} (**path a**). During this process, the chiral phosphate could control the facial selectivity of such reaction via both hydrogen-bonding interactions with the N-H

bond adjacent to the aryl group and ion-pairing interactions in a concerted transition state. Then, reductive elimination of the resulting Cu(III) species **E** would afford the final product **3** along with the regeneration of the copper Cu(I) and the phosphoric acid. However, another pathway (**path b**) via single-electron oxidation of intermediate **D** to the corresponding carbocation intermediate **F**, which undergoes C-N bond formation to give final product **3**, could not be excluded at the present stage. Therefore, rigorous investigations are necessary to unambiguously elucidate the exact mechanism.”

1-(3,5-bis(trifluoromethyl)phenyl)-3-(4-chloro-6,6,7,7,8,8,9,9,9-nonafluoro-2,2-dimethylnonyl)urea (3U)

$^1\text{H NMR}$ (500 MHz, CDCl_3) δ 7.85 (s, 1H), 7.75 (s, 2H), 7.43 (s, 1H), 5.66 (t, $J = 6.0$ Hz, 1H), 4.33-4.28 (m, 1H), 3.32 (dd, $J = 14.0, 7.5$ Hz, 1H), 3.13 (dd, $J = 14.0, 5.5$ Hz, 1H), 2.67-2.39 (m, 2H), 1.90 (dd, $J = 15.5, 9.5$ Hz, 1H), 1.75-1.73 (m, 1H), 0.99 (s, 3H), 0.97 (s, 3H).

$^{13}\text{C NMR}$ (125 MHz, CDCl_3) δ 155.7, 140.3, 132.3 (q, $J = 33.4$ Hz), 123.03 (q, $J = 272.7$ Hz), 118.6, 116.0, 119.8-108.2 (m), 49.6, 49.3, 47.8, 40.8 (t, $J = 20.4$ Hz), 35.0, 25.9, 24.7.

$^{19}\text{F NMR}$ (376 MHz, CDCl_3) δ -63.5 (s, 6F), -81.3 (t, $J = 9.4$ Hz, 3F), -113.5 (s, 2F), -124.9 (s, 2F), -126.2 (s, 2F).

HRMS (ESI) m/z calcd. for $\text{C}_{20}\text{H}_{19}\text{ClF}_{15}\text{N}_2\text{O}$ $[\text{M}+\text{H}]^+$ 623.0941, found 623.0933.

1-(4-benzyl-4-chloro-6,6,7,7,8,8,9,9,9-nonafluoro-2,2-dimethylnonyl)-3-(3,5-bis(trifluoromethyl)phenyl)urea (3V)

¹H NMR (500 MHz, CDCl₃) δ 7.76 (s, 2H), 7.45 (s, 1H), 7.31 (s, 5H), 7.23 (s, 1H), 5.21 (t, *J* = 6.0 Hz, 1H), 3.37-3.30 (m, 3H), 3.21 (d, *J* = 14.0 Hz, 1H), 2.72-2.60 (m, 2H), 2.06 (s, 2H), 1.12 (s, 3H), 1.09 (s, 3H).

¹³C NMR (125 MHz, CDCl₃) δ 155.2, 140.5, 135.0, δ 132.2 (q, *J* = 33.3 Hz), 131.6, 128.1, 127.4, 123.1 (q, *J* = 272.7 Hz), 118.4, 115.8, 112.8, 118.4-106.2 (m), 71.6, 53.4, 50.4, 49.4, 47.7, 41.3 (t, *J* = 19.0 Hz), 36.7, 31.1, 27.5, 26.7.

¹⁹F NMR (376 MHz, CDCl₃) δ -63.4 (s, 6F), -81.2 (t, *J* = 9.5 Hz, 3F), -107.6 (AB, d, *J*_{F-F} = 271.5 Hz, 1F), -111.2 (AB, d, *J*_{F-F} = 274.1 Hz, 1F), -124.5 (s, 2F), -125.8 ~ -126.0 (m, 2F).

HRMS (ESI) *m/z* calcd. for C₂₇H₂₅ClF₁₅N₂O [M+H]⁺ 713.1410, found 713.1414.

Comment 5: Table 3. The word “difluoroacetylation” is not correct. It means addition of “CF₂HCO”. It may be “difluoro(methoxycarbonyl)methylation”? Please reconsider the name.

Our response: We really appreciate the referee for this valuable suggestion. According to your comment, we carefully checked the word “difluoroacetylation” and “difluoro(methoxycarbonyl)methylation” in the reported studies. The word *difluoroacetylation* has been changed to “difluoro(methoxycarbonyl)methylation” in the revised manuscript and Supplementary Information.

Comment 6: Page 9, line 8: a lower loading of the chiral catalyst (5 mol%) was emphasized, but Table 2 and 3 also use only 5 mol%. It is rather better to say “a higher loading (10 mol%) is required in case of difluoromethylation reaction”.

Our response: We thank the referee for bringing this issue to our attention and completely agree on your comment. We have changed “a lower loading of the chiral catalyst (*S*)-A1 (5 mol%)” to “in the present of 5 mol% of chiral catalyst (*S*)-A1”.

Comment 7: Page 12, line 11: The subsequent radical addition of Rf radical to... “radical” seems to be redundant.

Our response: We thank the referee for pointing out this issue. We have deleted the “radical” in the revised manuscript.

Comment 8: About the mechanism proposed, the Br anion stays on Cu(II)-CPA, but in the presence of Ag₂CO₃, Br anion could also be trapped by Ag. According to Table 1, CuBr, CuCl, CuI generally gave high ee, in contrast, CuOAc gave much lower ee. It may be because of the trap effect of halogen by Ag to help tight association with CPA? It is desirable to discuss on the effects of counter anion.

Our Response: We really appreciate the referee for this valuable suggestion and completely agree on your point. We have already discussed the effects of counter anion in the revised manuscript. “Initially, the R_f radical and chiral monophosphate or bisphosphate Cu(II) **B** or **B'** are generated from the single electron transfer reaction of the corresponding R_fSO_2Cl with CuBr and the phosphoric acid, along with the generation of a stoichiometric amount of sulfur dioxide and chloride anion. The chiral bisphosphate Cu(II) **B'** is possibly formed via halide abstraction from the monophosphate Cu(II) **B** with Ag_2CO_3 to help tight association of Cu with chiral counteranion³⁰. Here, Ag_2CO_3 acts as a chloride scavenger via the formation of insoluble AgCl in organic solution, thereby minimizing the strong acid HCl-associated unselective background reactions.” has been added and discussed in the revised manuscript.

Referee 1

Comment 1: *Selective introduction of the fluorine-containing groups into organic compounds is an important issue in organic synthesis. Prof. Xin-Yuan Liu and co-authors developed a method for asymmetric radical amino-fluoroalkylation in the presence of the chiral catalyst. The authors insist on the novelty of asymmetric addition of fluoroalkyl radicals. However, they ignored the related report (by Prof. Miyabe) involving asymmetric fluoroalkyl radical-addition (JOC 2012, 77, 8588).*

Our response: We very much appreciate this referee for considering that we have developed a method for asymmetric radical amino-fluoroalkylation in the presence of the chiral catalyst. However, this referee pointed out that we ignored the related paper (by Prof. Miyabe) involving asymmetric fluoroalkyl radical-addition (JOC **2012**, 77, 8588). Actually, in this work, Miyabe and co-workers reported perfluoroalkyl radical addition to **electron-deficient alkenes** with excess amounts of Et₃B as the initiator by employing a chiral Lewis acid from chiral box ligand and Zn(OTf)₂ to activate amide group to realize enantioinduction. This reaction gave a mixture of *cis*-**B**, *trans*-**B** and **C** with moderate chemoselectivity, diastereoselectivity, and moderate to good enantioselectivity (Figure 1, eq. 1). Please kindly note that this reaction is completely different to our present work from the point of the reaction mechanism, the asymmetric catalytic system, the substrates, and the obtained products. In our present work, we describe our efforts toward the development of the dual Cu(I)/chiral phosphoric acid-catalyzed asymmetric radical intramolecular aminofluoroalkylation of **unactivated alkenes** with various commercially available fluoroalkyl sulfonyl chlorides in excellent yields (up 98% yield) and with excellent enantioselectivity (up to >99%) (Figure 2, eq. 2). This approach offers an unprecedented sustainable and broadly applicable platform enabling efficient access to four types of enantioenriched functionalized α -tertiary pyrrolidines bearing versatile β -fluoroalkyl groups with excellent efficiency, remarkable enantioselectivity, excellent functional group tolerance, and mild reaction conditions. Furthermore, this transformation enables the efficient construction of other useful chiral fluoroalkyl-containing building blocks. **Overall, we have successfully developed a strategically novel asymmetric radical approach for diversity-oriented synthesis of fundamental yet synthetically formidable enantioenriched fluorine-containing compounds.** Of course, according to this referee's comment, we have already added this reference as ref. 19 in the revised manuscript.

Figure 1

Comment 2: Furthermore, all the reactions in this report are too particular and tacky; the substrate scope is narrow. Only the specialized substrates are applicable and the products are not of general interest to readers of *Nature Communications*.

Our response: We do not agree that substrate scope is narrow and the products are not of general interest. At first, in the present work, it should be noted that the present radical process is a rather general reaction that can be extended to a wide range of substrate scope including alkenyl ureas with excellent functionality tolerance (**20 examples with various group-tethered and nonbranched substrates**), and **four types** of commercially available fluoroalkyl sulfonyl chlorides ($n\text{-C}_4\text{F}_9\text{SO}_2\text{Cl}$, $\text{MeO}_2\text{CCF}_2\text{SO}_2\text{Cl}$, $\text{HCF}_2\text{SO}_2\text{Cl}$, $\text{CF}_3\text{SO}_2\text{Cl}$). Therefore, this approach offers a sustainable and broadly applicable platform enabling efficient access to four types of enantioenriched functionalized α -tertiary pyrrolidines bearing versatile β -fluoroalkyl groups for **50** examples in excellent yields (up 98% yield) and with excellent enantioselectivity (up to >99%) with excellent efficiency, remarkable enantioselectivity, excellent functional group tolerance, and mild reaction conditions (Figure 2, eq. 2). Second, according to the third referee's valuable suggestion, we have made efforts to investigate the intramolecular trifluoromethylation of the unbranched substrate **1s** using the current method. It is more encouraging to note that the unbranched substrate **1s** underwent the current aminotrifluoromethylation reaction smoothly to deliver the desired product **6H** in 60% yield with 91% ee, which showed dramatically enhanced reactivity as compared with that using Togni's reagent as the CF_3 source to give low product yield and ee in case of such a substrate (also see the response to Comment 3 of Referee 3.). These results indicated that

trifluoromethanesulfonyl chloride as the CF_3 source exhibited a number of clear advantages in terms of high reactivity, low-cost and simple work-up over its Togni's reagent-based counterpart, rendering the current reaction system a more appealing alternative to the previous approach. Third, this transformation enables the efficient construction of other useful chiral fluoroalkyl-containing building blocks (see Figure 2 in the revised manuscript). Finally, chiral fluorinated amines bearing fluoroalkyl groups, such as perfluoroalkyl, trifluoromethyl, and difluoromethyl groups, have been gaining increasing interest among medicinal chemists as important synthetic building blocks in the design of pharmaceuticals and agrochemicals because these moieties can favorably affect the physical and biological properties of compounds. In particular, chiral amines containing a difluoromethyl group (CF_2H), which could act as lipophilic hydrogen bond donors and as bioisosteres of alcohols and thiols, should be of great importance for medicinal chemistry. Thus, the synthesis of optically pure amines containing various fluoroalkyl groups is very important for the organic synthesis (see refs. 1-9 in the revised manuscript). In the present work, we have provided a sustainable and broadly applicable platform enabling efficient access to four types of enantioenriched functionalized α -tertiary pyrrolidines bearing versatile β -fluoroalkyl groups in excellent yields (up 98% yield) and with excellent enantioselectivity (up to >99%). ***Overall, we have successfully developed a strategically novel asymmetric radical approach for diversity-oriented synthesis of fundamental yet synthetically formidable enantioenriched fluorine-containing compounds, thus reflecting the important synthetic utility of this method in synthetic and medicinal chemistry related fields.***

After reorganizing this manuscript carefully including a number of additional novel results according to all referees' valuable comments and given the comments of Referees 2 and 3 that "*This is a high-quality paper on the enantioselective radical aminoperfluoroalkylation of alkenes. The authors have done a very good job in developing this asymmetric process, demonstrating good scope and excellent enantioselectivities. The SI is also done very well. I very much enjoyed reading this paper and recommend it for publication in Nature Comm.*" and "*This manuscript describes an asymmetric aminoperfluoroalkylation of alkenes with fluoroalkylated sulfonyl chlorides by using the combination of copper catalyst and chiral phosphoric acid. Although the catalytic system was almost the same as the previous asymmetric aminotrifluoromethylation with Togni's reagent reported by the same authors (ref. 33), this method can provide highly optically active pyrrolidines with not only CF_3 but also C_4F_9 , CF_2H , and CF_2COOMe by using fluoroalkyl sulfonyl chlorides as the fluoroalkyl source. Although the concept for the asymmetric induction is same as previously reported reaction using Togni's Reagent, the variation of commercially available fluoroalkyl sulfonyl chloride is much wider and the reaction conditions reported here could be useful and versatile. Therefore this reviewer support this work to be publishable in Nature Commun*", we believe that the revised manuscript should meet the criteria of innovation and significance for publication in *Nature Communications*.

REVIEWERS' COMMENTS:

Reviewer #3 (Remarks to the Author):

The manuscript has been improved by the correction and addition of new data/discussion. Therefore the revised manuscript would be publishable.

Comments of Reviewer 3:

The manuscript has been improved by the correction and addition of new data/discussion. Therefore the revised manuscript would be publishable.

Our response:

We sincerely thank the reviewer for recommendation of publishing our paper on Nature Communications and appreciate the valuable suggestions from the reviewer.